

# 1 On the set of deterministic phenomena preceding the earthquake June 2 25, 2021 with a magnitude of 5.4 near the city of Yayladere (Turkey)

Alexandr Volvach[1], Lev Kogan[2], Konstantin Kanonidi[3], Igor Bubukin[4], Valeria Shtenberg[2],
Larisa Volvach[1]
[1]Radio Astronomy and Geodynamics Department of Crimean Astrophysical Observatory, Katsively, RT-22 Crimea, Ukraine
[2]Nizhny Novgorod State University of Architecture and Civil Engineering, Nizhny Novgorod, Russia
[3]Pushkov Institute of Terrestrial Magnetism, Ionosphere and Radiowave Propagation of RAS, Moscow, Russia
[4]Radiophysical Research Institute of N.I. Lobachevsky Nizhny Novgorod State University, Nizhny Novgorod, Russia
*Correspondence to*: Alexandr E. Volvach (volvach@meta.ua)
**Abstract.** The article examines the 5.4 magnitude earthquake that occurred on June 25, 2021 in the vicinity of Yayladere
(Turkey). The analysis of the geomagnetic measurements carried out revealed a set of deterministic processes that preceded
this event and are interpreted as its precursors. An estimate is made of the average time between the interval of existence of
such phenomena and the moment of the earthquake under consideration. As a result, close values of the corresponding averaged
time intervals were obtained for the statistics of all three components of the geomagnetic field considered in the article. The
proposed technique can be used to predict seismic processes in various regions of the world in a near real-time mode.

## 1 Introduction

During the last years a number of scientists published the results of researches, indicative of possibility of registration of
harbingers of strong earthquakes in the distance of more than 5000 km, and in some cases more than 10 000 km (Hasanov &
Keramova 2006; Lyubushin 2008; Sobolev et al. 2008; Khain and Khalilov 2008).
Philosophy of short-term forecasting of earthquakes hasn't undergone essential changes during the whole history of its
presence. The basis of all technologies of short-term forecasting the earthquakes is to create the network of stations, which
register the changes of geophysical, geochemical, hydro-geological and other parameters of geological medium before strong
earthquakes near potential sources of possible earthquakes.
This article discusses deterministic processes that were detected in the last few hours before an event with a magnitude of M
= 5.4 with an epicenter of 39.196° N, 40.165° E, which occurred at 18:28:37 (UTC) on 25.06.2021, 8 kilometers from the city
of Yayladere (Turkey). The object of the study was the statistics of magnetic field measurements carried out at the test site of
the Radio Astronomy and Geodynamics Department of the Crimean Astrophysical Observatory in Katsiveli (Crimea) as well
as at the Borok station during the period 00:00:00 on 24.06.2021 – 23:59:59 on 25.06.2021. In order to search for precursors
of an impending earthquake, the technique proposed in (Volvach et al. 2022a,b,c,d; Volvach et al. 2023; Kogan et al. 2021;
Kogan 2015)) and based on the change in the properties of the probability density of any random process when even a very



small term appears in its composition, the nature of which is weakly related to a set of commonly occurring phenomena, was
applied.
As will be shown below, the proposed approach made it possible to detect a number of phenomena that indicate a high
probability of an imminent seismic event of significant magnitude. Thus, statistical methods are applicable not only within the
framework of the Gutenberg-Richter theory (Gutenberg & Richter 1956; Amitrano 2012; Sanchez & Vega-Jorquera 2018),
which makes it possible to assess the possibility of earthquakes of a given magnitude on medium-term, on the order of months
and years, of time intervals, but also in relation to small, up to several hours, intervals before the onset of impending seismic
events, which is close to the current time regime.

## 2 Mathematical apparatus used in solving the objective

The article introduces the assumption used in (Volvach et al. 2022a,b,c; Kogan et al. 2021; Kogan 2015) that any physical
fields $x(t)$ measured in seismically active regions can be written in the form
$$x(t) = x_1(t) + x_2(t). \tag{1}$$

Here $x_1(t)$ is the background noise associated with the influence of a set of ordinary phenomena and processes. Whereas the
term $x_2(t)$ is solely due to the impact of an impending seismic event. At the same time, the hypothesis about the weak statistical
dependence of these two random variables (RV) is introduced into consideration.
As indicated in the Introduction, below we will consider the properties of magnetic field statistics. In this case, the entire
measurement period is divided into implementation segments of one minute duration, and each such segment is assigned 60
measuring counts (one per second). As in (Volvach et al. 2022), to each implementation segment is matched value of the
statistical functional
$$L(n) = \frac{A}{M} \sum_{l=n-(M-1)}^{n} |\mathcal{L}_l|, \quad \mathcal{L}_l = \sum_{m=0}^{N-1} (-1)^m P_{m,l}. \tag{2}$$

Here, $M = 100$, the factor $A = 1000$ is necessary to obtain a convenient for analysis range of values $L(n)$, $l$ is the number of
the implementation segment, the value $n$ corresponds to the moment of observation, that is, the completion time of the
implementation segment with this number, and $P_{m,l}$ is the probability of the measured value of the function $f[x(t)]$,
corresponding to the indicated segment number $l$, falls into the $m$-th interval of the range of values of the form
$$f[x(t)]_{min} + mh \le f[x(t)] < f[x(t)]_{min} + (m+1)h, \qquad 0 \le m \le N-1 \tag{3}$$

In (3) $f[X] = \sin(X)$ (see[2]) and $f[x(t)]_{min} \ge -1$ is the smallest value $f[x(t)]$ for a given implementation segment. (The
$x(t)$ values were measured and accepted in calculations in nT units.) In addition, in (3) the sampling interval is set equal to $h =$
0.1, therefore, in (2) and (3) $N = 20$. According to (Volvach et al. 2022a,b,c; Kogan et al. 2021; Kogan 2015), functional (2)
essentially depends on the level of entropy of the values of the random process, the value of which significantly increases
when an independent stochastic term $x_2(t)$ appears in (1).



**3 Data and analysis of the properties of the functional L(N) constructed from the measurements of the magnetic field**

The International Real-time Magnetic Observatory Network - the global network of observatories, monitoring the Earth's magnetic field and the data are available as digital data files (www.intermagnet.org, ftp.seismo.nrcan.gc.ca).

Fig. 1 shows the dependence $L(n)$, constructed from measurements of the H-component of the magnetic field on June 24 and 25, 2021, with a magnetometer at the above-mentioned test site of the CrAO in Katsiveli. Hereinafter, the horizontal axis corresponds to UTC time in minutes starting from $n = M = 100 \, min$ after the first moment of the first day of the considered 48-hour time interval (see (2)). In all figures, the solid vertical red straight line (drawn at $n = 2459 \, min$) denotes the moment of the earthquake under consideration. Let us introduce a number of definitions, illustrating them with examples from Fig. 1.

1. We call the main extrema the set of points of maxima and minima of the dependence $L(n)$, which on the intervals of both increasing and decreasing corresponds to the variation $\delta L$ of the values of this functional, satisfying the condition

$$\delta L \geq 0.2 \, \Delta L_{max} . \tag{4}$$

In (4), $\Delta L_{max}$ is the difference between the largest maximum and the smallest minimum of functional (2) from the start to the moment of the considered earthquake. Here and below, unless the opposite condition is set, small-scale variations in the values of $L(n)$ are not taken into account. The name "small-scale fluctuations" will denote fluctuations that are smaller in magnitude than the right-hand side of (4) and, at the same time, are small in amplitude and duration compared to the corresponding extremum. In this case, we assume that in the region of the corresponding extremum, condition (4) is met both in the region of increasing and in the interval of decreasing the $L(n)$ curve. In addition, within the limits of each such section, there should be a quasi-constant rate of change of this dependence. In Fig. 1, the main extrema, in particular, are the points $h, a, b, c, d, l, f, e, g, p$.

2. We will call the sections of the dependence $L(n)$, which have the properties of quasi-monotonicity and quasi-constant rate of change (also neglecting small-scale variations), as local trends. We assume that they correspond to the $\delta L$ variation satisfying (4). In this case, the geometric deviation of the points of the local trend from the segment of the straight line connecting the points of its beginning and end should not exceed 15% of the modulus of the difference between the values of the dependence $L(n)$ at these points. We emphasize that for the main extremums defined above, the corresponding intervals of increase and decrease can also be considered as local trends. Examples of local trends in Fig. 1 are, for example, segments $a - b$, $i - j$ and $e - u$ of a given curve.

3. Let us denote as global extrema those points of maxima and minima of the dependence $L(n)$, which correspond to the largest or, correspondingly, the smallest values of it over an interval of at least 24 hours. Moreover, they should be located near the middle of such a time interval. Such a point is, for example, point $q$, see Fig. 1.

4. Let's call the set of the main and global extrema, as well as the boundary points of trends, the guide points.

5. We will call a channel a collection of two straight lines, each of which is drawn through two guide points, provided that the difference $\Delta \alpha$ of the directions of these straight lines satisfies the constraint

$$\Delta \alpha \leq 1.5° . \tag{5}$$





Here and below, we assume that the angle $\Delta\alpha$ corresponds to the condition of equality of the geometric dimensions of the
length units of the vertical and horizontal axes. At the same time, in this article, in all the figures, the unit of the abscissa axis
is slightly greater than the unit of the ordinate axis. he angle $\Delta\tilde{\alpha}$ corresponding to these figures between the directions of the
"shores" of any of the channels is related to $\Delta\alpha$ by the ratio

99                  $\Delta\alpha \leq 0.8\,\Delta\tilde{\alpha}.$                                            (6)

The duration of the channel's existence, that is, the time interval between the guide points through which its boundaries pass,
must be at least 150 minutes.
For clarity, all the figures show straight lines strictly parallel to the channel boundaries. They have the same number as the
corresponding border, but with an additional index «′» (for example, in Fig. 1, straight line *1′* is assigned a parallel straight *1*).
In order to achieve a low probability of random proximity of the directions of the channel boundaries, we introduce additional
conditions. They are reduced either to the existence of one more (besides the channel boundaries) straight line satisfying (5)
as applied to at least one of the channel banks, or to a significant overlap area (with respect to the abscissa axis) of the areas
between the guide points through which these boundaries pass, or to the commensurability of the time intervals between both
pairs of such points with half a day, or to a small (compared to the duration of the channel's existence) distance between the
points of the beginning of the boundaries of the channels and / or between the points of their termination, etc.
In Fig. 1 for the channel created by straight lines *1–2* (hereinafter we will introduce the designations of the form "channel *1–*
*2*") the angle $\Delta\tilde{\alpha} \ll 1.0°$, and for channels *3–4* and *5–6* the value $\Delta\tilde{\alpha} \approx 1.2°$ and, accordingly, $\Delta\tilde{\alpha} \approx 1.0°$. Taking into account
(6), from here, for the angles $\Delta\alpha$ corresponding to these channels, in this figure it follows that (5) is obviously fulfilled. Pairs
of straight lines *1–2* and *3–4* are examples of spacing channels, and *5–6* is channel with substantially overlapping boundaries.
Straight *1* and *4* are drawn through the guide points $q$, $j$ and, respectively, $i$, $d$; for the rest of the channel boundaries, the
corresponding guide points are seen from Fig. 1.
6. We will call a sliding boundary (SB) any straight line that is drawn through two guide points and is tested by the curve $L(n)$
at two more points, one of which is also a guide (we will call it additional). The term "testing" means either a simple intersection
of the curve (2) and a given straight line, or its passage from the guiding point at such a close distance that the deviation
coefficient $\delta$ satisfies the condition

120                  $\delta \leq 1.6\%,$                                              (7)

where $\delta = \frac{\Delta S}{L_g(n)} \cdot 100\%$. Here $\Delta S$ is the minimum vertical distance on the plane $\{n, L(n)\}$ of the Cartesian variable between
the straight line under consideration (both the SB and the channel boundary) and the given guide point, and $L_g(n)$ is the value
of the functional $L(n)$ in this point.
An additional point can be located either between the two guides through which the SB line passes, or after them, but until the
fourth test. In Fig. 1, the sliding border is straight line *1* drawn through the global extremum $q$ and the guide point $j$. Point m
is additional. Her area is highlighted with an orange ellipse. In this case, the coefficient $\delta \approx 0.95\%$ satisfies (7).



7. We also define that if in the region of small-scale fluctuations there are more than three approximations of the curve $L(n)$
to the corresponding straight line when fulfilling (5), then any such segment of this curve will be assumed to correspond to
one test.
Note that in Fig. 1, the variation value for the $w - t$ segment of the $L(n)$ curve is less than the right-hand side of (4) and
therefore does not satisfy the definition of the local trend. It is also obvious that w is not the main or global extremum. In
addition, at this point and its vicinity, the intersection of the curve under consideration and straight *1* does not occur. Therefore,
despite the fulfillment of condition (7) for $w$ (as applied to straight *1*), the point $w$ is not included in the number of testing
points.
Thus, straight line *1* is both the SB and the bank of the channel at the same time. Hereinafter, those channel boundaries that
are not sliding boundaries will be denoted by dashed straight lines, and the SB themselves will be denoted by solid straight
lines.
With regard to channel boundaries, an additional point, if any, can also be located either between those two guide points
through which this boundary passes, or after them. In this case, for one channel boundary, there can be no more than three
consecutive tests. If this rule is violated, any segment of the corresponding straight line containing the indicated three
sequentially tested points is considered the channel boundary. (These three points include also and those two guide points
through which the given channel boundary is drawn.)
We now turn directly to the search for the precursors of the earthquake under consideration. In Fig. 1, three vertical dashed
straight lines in the right part of it are drawn from the points of the fifth testing of the curve $L(n)$ of the bank lines of channels
*1–2*, *3–4*, and *5–6*. (Here and below, similar vertical dashed straight lines mark the moments of time corresponding to the
appearance of the "graphic precursors" considered below). In this case, the lower boundary of the first of these channels is
assumed to be the segment $q - j$ of straight line *1* containing an additional point $m$. The fact of the existence of such channels
can be interpreted as the emergence of a set of deterministic processes immediately before an earthquake.
The time intervals from the indicated points of the fifth testing to the moment of the earthquake are $T_{H,1} = 296\ min$, $T_{H,2} =$
$278\ min$ and $T_{H,3} = 197\ min$. As will be shown below, the phenomena of the fifth testing of the channel boundaries precede
the considered seismic event and when carrying out a similar statistical analysis for the E- and Z- components of the measured
geomagnetic field. Therefore, we will consider the moment of the indicated fifth testing as the time of realization of one of the
types of "graphic precursors" of an impending earthquake.
In addition, as will be confirmed below, in the study of all three components of the magnetic field, the point of the fourth
testing of the sliding boundary from the side of the curve $L(n)$ is also a recurring precursor of an earthquake. This effect takes
place in Fig. 1 at point $v$ in relation to sliding border in the form of straight line *1*. (At this point, the indicated fourth testing
of this SB takes place, also marked with a vertical dotted straight line, see Fig. 1.) The appearance of SB can also be called the
emergence of a deterministic phenomenon a few hours before the impending earthquake.
Let us introduce the condition that hereinafter we take into account only the precursors arising in no more than a time



$$T_{max} = 720 \, min \tag{8}$$

before the start of the earthquake. Therefore, in this case, the time interval from point $v$ in Fig. 1 until the moment of the
seismic event is not taken into account.
In Fig. 2 shows the dependence $L(n)$ for the E-component of the magnetic field (data from the magnetometer in Katsiveli).
Here, channels $1$–$2$, $3$–$4$ and $5$–$6$ are drawn, for which the angles $\Delta\alpha < \Delta\widetilde{\alpha} \ll 1°$ (see (5) and (6)). Segments $a - b$, $i - j$,
$c - d$, $e - f$, $m - l$ and $l - h$ (their boundaries contain all but $q$, the guide points through which all the indicated straight are
drawn) satisfy the definition of local trends. For points $q$ and $e$, the definitions of the main and, accordingly, global extremum
are fulfilled.
In the region of small-scale fluctuations, marked with an orange ellipse, there are more than three approximations of the curve
$L(n)$ to straight line $5$ when (5) is fulfilled. According to definition VII introduced above, we consider that such a segment of
this curve corresponds to one test. Taking this into account, we find that the specified straight is a sliding boundary, the fourth
testing of which takes place at point $g$, see Fig. 2.
By analogy with Fig. 1, we assume that the points of the fifth testing of the channel boundaries of the curve (2) or the fourth
testing of the line of the specified SB correspond to the time of realization of the "graphic precursors" of an impending seismic
event. Therefore, also by analogy with Fig. 1, we postpone the intervals $T_{E,1} = 434 \, min$, $T_{E,2} = 251 \, min$, $T_{E,3} = 231 \, min$
and $T_{E,4} = 181 \, min$ from the moments of the appearance of such precursors to the time of the beginning of this earthquake.
As in the previous case, in Fig. 2 these intervals correspond to green horizontal lines.
In Fig. 3 shows the dependence $L(n)$ for the Z-component of the magnetic field (data from the magnetometer in Katsiveli).
Here, channels $1$–$2$ are drawn (here $\Delta\widetilde{\alpha} \approx 0.8°$), as well as $2$–$3$ and $4$–$5$, for which the angles $\Delta\widetilde{\alpha} \ll 1°$ (see (5) and (6)). Note
that straights $1$, $2$ and $3$ are almost parallel. Sliding boundaries $6$, $7$ and $8$ are drawn through the first and third (counting from
the left) test points; areas of additional points are marked with ellipses. The ellipse also marks an additional point $c$ for line $4$,
which is the lower boundary of channel $4$–$5$ (this guide point is also the beginning of SG 8). Here, the guide point $a$, through
which this line passes, is the beginning of the local trend $a - b$. For SB $7$, the region of small fluctuations within the
corresponding ellipse is associated with one test. Taking into account (8) for SB $6$ and $7$, the intervals from the time of the
fourth testing to the moment of the earthquake are not considered. Intervals $T_{Z,1} = 596 \, min$, $T_{Z,2} = 588 \, min$, $T_{Z,3} =$
$324 \, min$, $T_{Z,4} = 290 \, min$ and $T_{Z,5} = 126 \, min$ depicted by green horizontal lines.
In order to verify the results obtained in Fig. 4 shows the dependence $L(n)$, which corresponds to the measurements of the Z-
component of the magnetic field, carried out on June 24 and 25, 2021 on the Borok magnetometer. For channels каналов $1$–
$2$, $3$–$4$, $5$–$6$ and $7$–$8$ the angles $\Delta\widetilde{\alpha}$ satisfy (5) (the degree of parallelism of their boundaries is illustrated by pairs of straight
lines $1'$ and $2'$, $3'$ and $4'$, $5'$ and $6'$ as well as $7'$ and $8'$, see Fig. 4. Moving boundaries $9$ are drawn through the first and third
(counting from the left) tested points. The areas of additional points are marked with ellipses. Point a can be considered
simultaneously the point of the fifth testing for channel $5$–$6$ and fourth testing for the sliding border $9$. Intervals $T'_{Z,1} =$
$569 \, min$, $T'_{Z,2} = 241 \, min$, $T'_{Z,3} = 122 \, min$, $T'_{Z,4} = 122 \, min$ and $T'_{Z,5} = 27 \, min$ are shown with green horizontal lines.





The range of values of these intervals, as well as their average values, are comparable with the analogous parameters for Fig.
194 3.

For ease of comparison in Table 1 shows all the obtained values of the intervals $T_H$, $T_E$ and $T_Z$ (first 5 lines), as well as their
average values of the form $\langle...\rangle_{720}$ and $\langle...\rangle_{360}$ over the last 720 and, respectively, 360 minutes before the earthquake.

## 4 Discussion of the results

The results presented in this paper rely on data collected at magnetic observatories. We used the INTERMAGNET stations -
the global network of observatories, monitoring the Earth's magnetic field (www.intermagnet.org).
As a result of analyzing the statistics of magnetic field measurements carried out in the last two days before the earthquake,
which occurred at 18:28:37 (UTC) on June 25, 2021, 8 kilometers from the city of Yayladere (Turkey), we come to the
following conclusions.
1. In the last hours before a given seismic event, there is an effect of concentration of recurring deterministic phenomena
identified in the previous section of the article and interpreted as a kind of "graphic precursors" of an impending seismic event.
In this case, for all three components of the geomagnetic field, a kind of "ladder of precursors" appears, see Fig. 1 – 3.
2. The most common precursor is the phenomenon of fivefold dependency testing (2) of the boundaries of the channels defined
above. These channels are formed by a pair of almost parallel straight lines (see (5)), which limit the values of the $L(n)$ curve
for at least 150 minutes.
3. Sliding boundaries in the form of straight lines, each of which is tested at least four times by the dependence in question, is
also an essential type of "graphical precursor". In this case, the first three tests of this kind must correspond to topologically
selected points of the curve (2).
4. As follows from the analysis of the data in Table 1, in the last 6 hours before the event, the values of the intervals $T_H$, $T_E$
and $T_Z$ are comparable. In particular, this manifests itself in the relative closeness of the average values $\langle T_H \rangle_{360}$, $\langle T_E \rangle_{360}$ and
$\langle T_Z \rangle_{360}$, which correspond to averaging over the last 360 minutes before the onset of this earthquake. As the authors believe,
this fact is an argument in favor of the reliability of the proposed method.
5. The discovered effects of the existence of deterministic phenomena in the form of channels and sliding boundaries are quite
reliable (outside the scope of this work, they were tested on a large array of data on various earthquakes and corresponding
measurements of fields of a very different nature), but at the same time they are difficult to explain. It is possible that the
existence of such straight linear boundaries of the values of functional (2) is associated with an almost constant (on a 48-hour
scale) speed of movement of lithospheric plates; this assumption is completely hypothetical. In addition, significant difficulties
in explaining the discussed phenomena can be associated with poor theoretical knowledge of the area of probability theory
corresponding to the phenomena under consideration, which can be attributed to "anti-Gaussian" random processes associated
with the properties of the sum of a large number of strongly dependent random variables.
6. When calculating according to the Borok magnetometer data for the Z-component of the magnetic field, the range of time
intervals from the appearance of "graphic" precursors to the beginning of the earthquake under consideration turns out to be



commensurate with the analogous parameters for the magnetometer data at the Katsiveli test site. This fact testifies both to the
stability of the applied technique to variations in the initial data, and to the reliability of the operation of the equipment in both
equipment complexes.
The study of the influence of cracks on various physical properties of rocks is one of the main tasks of earthquake prediction.
The conductivity of natural media is due to the transfer of electric charges by the through current of electrons, ions, holes. In
the epicenter of an earthquake, in addition to an increase in the number and length of elementary cracks, their closure and the
appearance of a main crack, there is also a change in the electrical resistance of rocks. When the soil level and density change,
the specific electrical conductivity of rocks changes by several orders of magnitude compared to the initial value, which leads
to a change in the characteristics of the magnetic field.
With the help of the INTERMAGNET international network, the Earth's magnetic field is monitored, which makes it possible
to create a technique that can be used to predict seismic processes in various regions of the world in close to real time.
**5 Conclusions**
In the article, based on the analysis of the results of measurements of the geomagnetic field, the deterministic phenomena
associated with the process of "final preparation" of the earthquake with a magnitude of 5.4, which occurred on June 25, 2021,
near the Turkish city of Yayladere, were considered. Based on the results obtained, "graphic precursors" were identified in the
form of repeating phenomena of the emergence of channels and sliding boundaries, the properties of which were investigated
in this work. The authors consider the most promising direction of further research to determine, within the framework of the
methodology proposed in the article, not only the time, but also the coordinates of the future epicenter, and, possibly, the
magnitude of the expected earthquake.
**Data availability statement**
The data underlying this paper are available in the paper and through the electronic resources: www.intermagnet.org,
ftp.seismo.nrcan.gc.ca.
**Acknowledgments**
The results presented in this paper rely on data collected at magnetic observatories. We thank the national institutes that support
them and INTERMAGNET for promoting high standards of magnetic observatory practice (www.intermagnet.org).



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













**Table 1. Values of intervals from the time of realization of «graphic precursors» to the moment of the beginning of the earthquake, as well as their averaging.**


| $I$ | $T_{\mathrm{H},i}$, $min$ | $T_{\mathrm{E},i}$, $min$ | $T_{\mathrm{Z},i}$, $min$ | $T'_{\mathrm{Z},i}$, $min$ |
|---|---|---|---|---|
| 1 | 296 | 434 | 596 | 569 |
| 2 | 278 | 251 | 588 | 241 |
| 3 | 197 | 231 | 324 | 122 |
| 4 | | 181 | 290 | 122 |
| 5 | | | 126 | 27 |
| $\langle\ldots\rangle_{720}$ | 257 | 274 | 384 | 216 |
| $\langle\ldots\rangle_{360}$ | 257 | 221 | 246 | 102 |























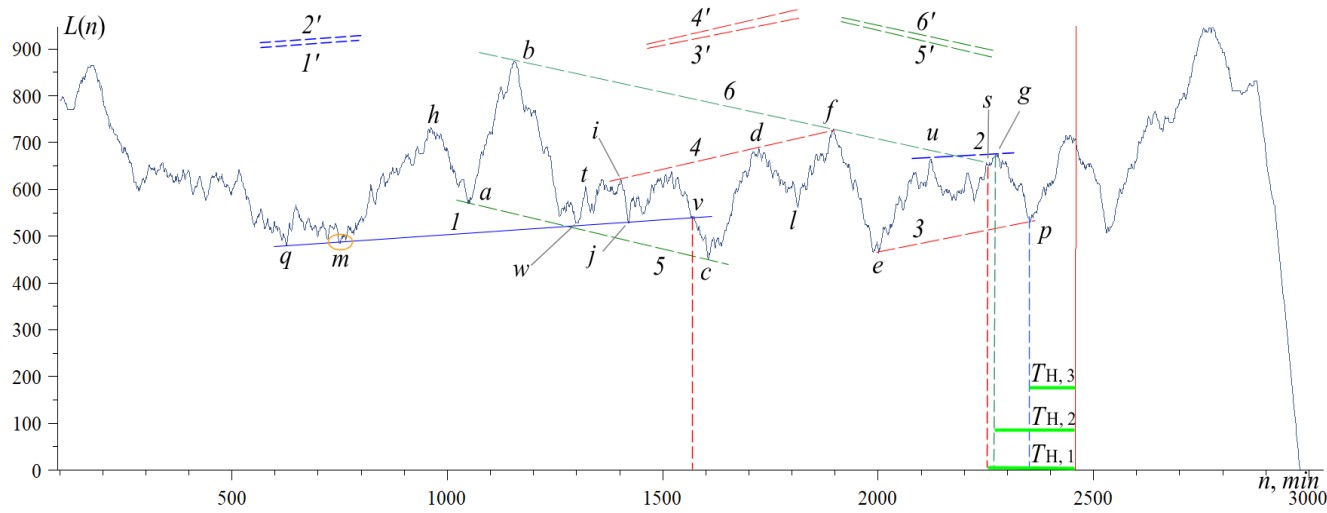


**Figure 1. Dependence L(n) for the H-component of the magnetic field from measurements at the Katsiveli.**







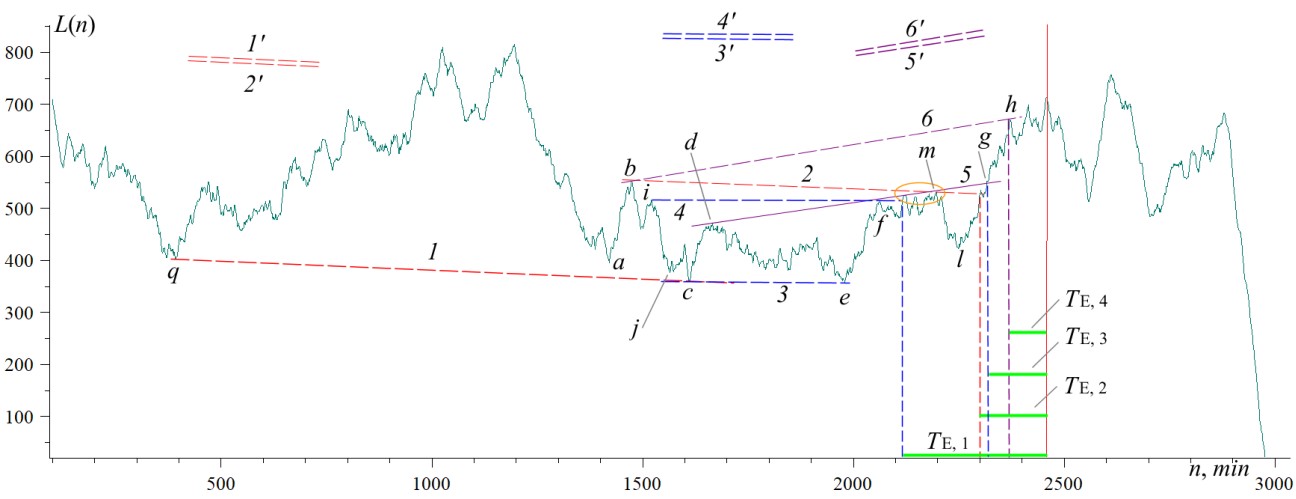

**Figure 2. Dependence L(n) for the E-component of the magnetic field from measurements at the Katsiveli.**

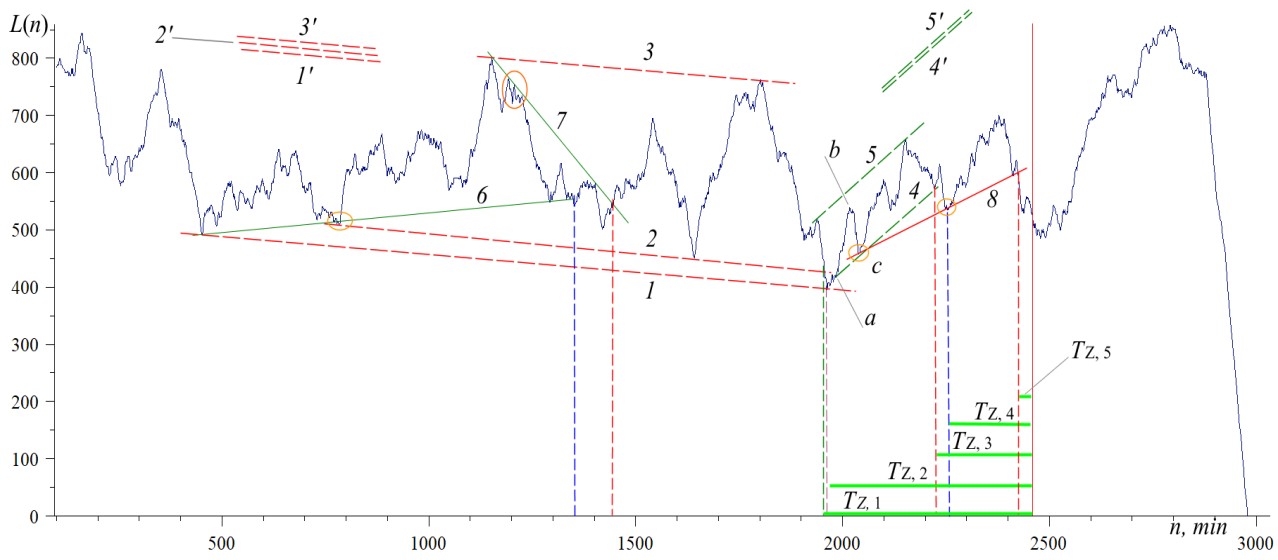

**Figure. 3. Dependence L(n) for the Z-component of the magnetic field from measurements at the Katsiveli.**





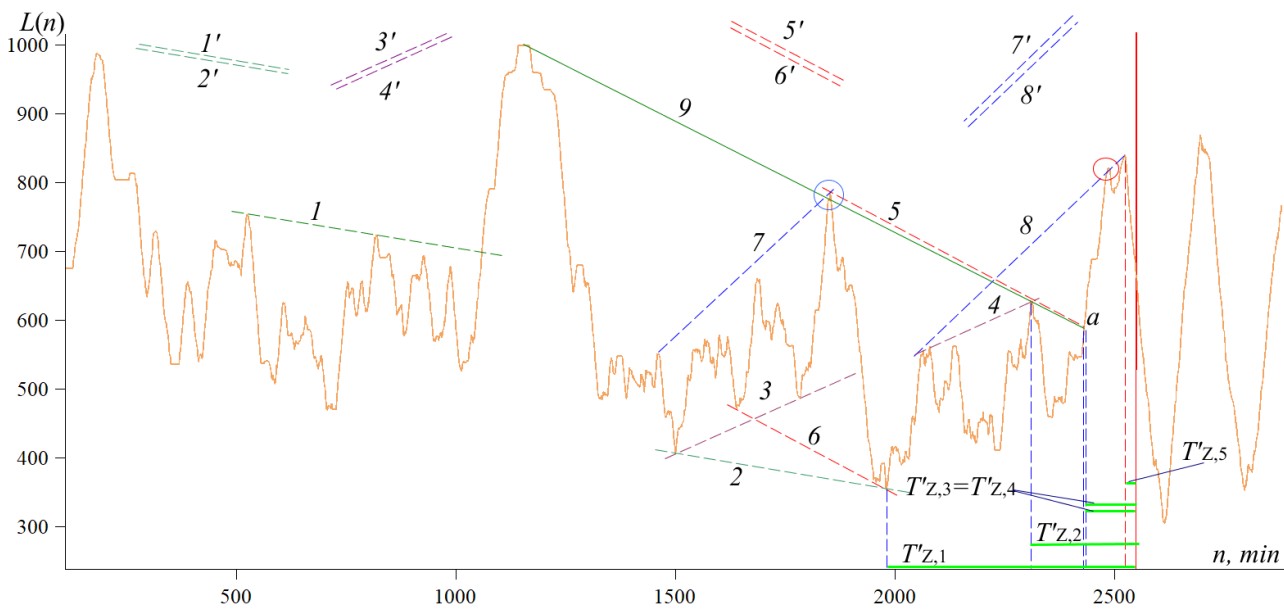

**Figure. 4. Dependence L(n) for the Z-component of the magnetic field from measurements at the Borok.**