# Peer review of "On the set of deterministic phenomena preceding the earthquake June 2 25, 2021 with a magnitude of 5.4 near the city of Yayladere (Turkey)"

_Annales Geophysicae, 2023_

## Author Comment (AC1)

**Dear Referee!**

We carefully considered all your comments and made adjustments to the text of the manuscript. Thank you for efforts to improve the quality of the manuscript.

All the best! The Authors.

**\*\*\*\*\*\**

• They affirmed that the analysis of the geomagnetic measurements revealed a set of deterministic processes that preceded the seismic event and they are interpreted as its precursors. However, the authors had never shown that these processes do not appear in intervals of time far away from the earthquake.

**Answer:**

In response to this reviewer's comment, the authors would like to refer to Volvach et al. 2022a,b,c; Volvach et al. 2023a. Within the framework of an approach similar to that used in this article, they consider precursors for 37 earthquakes. For 26 of them (with the exception of aftershocks that are not far removed in time from the previous event), almost all of the considered precursors are recorded before their "own" earthquake at a time interval of no more than 20–25% of the total considered measurement period  $\Delta t_{glob}$ . Moreover, in the absolute majority of the cases under consideration, this interval is on the order of 5–10% or less of  $\Delta t_{glob}$ . The same effect also takes place in relation to this article, which indicates the stability of the results of the applied methodology.

We also point out that Volvach et al. 2023b (it was published after this article was sent to the editors of the journal) considers the precursors of an earthquake of magnitude 5.3 (Metsavan, Armenia, 2022-02-13, 18:25:56 (UTC)). In this article, using a similar approach, the 12-day interval before the moment of the indicated earthquake was considered. At the same time, for 10 days before this event, no earthquakes of comparable magnitude were recorded in an area with a radius of about 2000 km from the epicenter point. As a result, the fact of a sharp increase in the concentration of the precursors under consideration as we approach the moment of the seismic event was proved. At the same time, the number of such precursors increases as the period of time considered before a given earthquake increases. In order to test the robustness of the approach used, Volvach et al. 2023b, significantly more stringent conditions were applied than in the peer-reviewed article and all other referenced works. With the same level of stringency as in this article, the number of precursors in Volvach et al. 2023b is getting significantly larger.

As an example, in response to the review Fig. 1a. It corresponds to Fig. 5 from Volvach et al. 2023b and built under the stringent conditions introduced in the cited paper. At the same time Fig. 1b is built according to the same data as Fig. 1a, but the channels and sliding boundaries are constructed under conditions similar to those introduced in this article and much more simplified compared to Volvach et al. 2023b. As follows from the comparison Fig. 1a and Fig. 1b, in the latter case, the number of registered precursors of an approaching earthquake increases from 5 to 10, that is, two times. At the same time, there is a pronounced "gravitation" of the moments of registration of such precursors by the time of the beginning of the earthquake. In both figures, these precursors are located within an interval of about five days before the considered earthquake and, as in the above works, their concentration increases significantly as the moment of the onset of the event is approached.

Fig. 1a. Dependence L(n) corresponding to measurements of the X-component of the geomagnetic field on the GLK magnetometer in the period from 00:00 February 1, 2022 to 23:59 February 15, 2022.

---

## Author Comment (AC2)

Dear Referee!

We carefully considered all your comments and made adjustments to the text of the manuscript. Thank you for efforts to improve the quality of the manuscript.

All the best!
The Authors.
* * *
- **They affirmed that the analysis of the geomagnetic measurements revealed a set of deterministic processes that preceded the seismic event and they are interpreted as its precursors. However, the authors had never shown that these processes do not appear in intervals of time far away from the earthquake.**

**Answer:**

In response to this reviewer's comment, the authors would like to refer to Volvach et al. 2022a,b,c; Volvach et al. 2023a. Within the framework of an approach similar to that used in this article, they consider precursors for 37 earthquakes. For 26 of them (with the exception of aftershocks that are not far removed in time from the previous event), almost all of the considered precursors are recorded before their "own" earthquake at a time interval of no more than 20–25% of the total considered measurement period Δt_glob. Moreover, in the absolute majority of the cases under consideration, this interval is on the order of 5–10% or less of $\Delta t_{glob}$. The same effect also takes place in relation to this article, which indicates the stability of the results of the applied methodology.

We also point out that Volvach et al. 2023b (it was published after this article was sent to the editors of the journal) considers the precursors of an earthquake of magnitude 5.3 (Metsavan, Armenia, 2022-02-13, 18:25:56 (UTC)). In this article, using a similar approach, the 12-day interval before the moment of the indicated earthquake was considered. At the same time, for 10 days before this event, no earthquakes of comparable magnitude were recorded in an area with a radius of about 2000 km from the epicenter point. As a result, the fact of a sharp increase in the concentration of the precursors under consideration as we approach the moment of the seismic event was proved. At the same time, the number of such precursors increases as the period of time considered before a given earthquake increases. In order to test the robustness of the approach used, Volvach et al. 2023b, significantly more stringent conditions were applied than in the peer-reviewed article and all other referenced works. With the same level of stringency as in this article, the number of precursors in Volvach et al. 2023b is getting significantly larger.

As an example, in response to the review Fig. 1a. It corresponds to Fig. 5 from Volvach et al. 2023b and built under the stringent conditions introduced in the cited paper. At the same time Fig. 1b is built according to the same data as Fig. 1a, but the channels and sliding boundaries are constructed under conditions similar to those introduced in this article and much more simplified compared to Volvach et al. 2023b. As follows from the comparison Fig. 1a and Fig. 1b, in the latter case, the number of registered precursors of an approaching earthquake increases from 5 to 10, that is, two times. At the same time, there is a pronounced "gravitation" of the moments of registration of such precursors by the time of the beginning of the earthquake. In both figures, these precursors are located within an interval of about five days before the considered earthquake and, as in the above works, their concentration increases significantly as the moment of the onset of the event is approached.

[Figure]

Fig. 1a. Dependence $L(n)$ corresponding to measurements of the X-component of the geomagnetic field on the GLK magnetometer in the period from 00:00 February 1, 2022 to 23:59 February 15, 2022.

[Figure]

Fig. 1b. Dependence $L(n)$ corresponding to measurements of the X-component of the geomagnetic field on the GLK magnetometer in the period from 00:00 February 1, 2022 to 23:59 February 15, 2022.

In this context, it should be pointed out that the effect of registering precursors in almost all cases over a time interval that is much shorter than the total measurement period under consideration - this phenomenon takes place both for relatively short, about 2 days (Volvach et al. 2022a,b; Volvach et al. 2023a) and much longer (Volvach et al. 2022c; Volvach et al. 2023b) measurement periods. (At the same time, the minimum values of the interval $\Delta n$ from the boundary of the local

trend to the nearest horizontal point of the curve $L(n)$ change proportionally to $\Delta t_{\text{glob}}$, that is, they are taken equal to at least 1.5% of the duration of this period of time (almost always the real values of $\Delta n$ turn out to be much larger); the value of the greatest variation $\Delta L$ of the values of this functional with a high probability also "automatically" increases in direct proportion to the total measurement time $\Delta t_{\text{glob}}$.)

Thus, the calculations carried out in this article, in comparison with the works listed above, confirm the conclusion that the forthcoming earthquake is preceded with a high probability by the occurrence of a series of statistical phenomena that are associated with the properties of the linear structures under consideration and are interpreted as precursors of an approaching seismic event. In this case, the set of moments of registration of precursors with a high probability tends to the time of the beginning of the upcoming earthquake. This fact means a high probability of connection of the considered statistical effects with the process of "final preparation" of the approaching earthquake.

The relevant text has been added to the "Discussion of the results" section of this article.

- **Lines 34-35: "the proposed approach made it possible to detect a number of phenomena that indicate a high probability of an imminent seismic event of significant magnitude". This probability is not calculated, neither there is a link with the magnitude of the earthquake.**
- **Answer:**

As follows from this article, as well as Volvach et al. 2022a,b; Volvach et al. 2023a, if at least 3 considered precursors occur within an interval of 300 minutes or more, according to the measurement statistics of at least one of the magnetic field components, the probability that an earthquake will occur within an interval of about an hour after the registration of the last specified precursor is very high. At the same time, this conclusion is obviously preliminary, subject to verification in the course of further studies and relating to the case of a total duration of the analyzed measurement period of the order of two days. The relevant text has been added to the "Discussion of the results" section of this article.

At the same time, of course, the reviewer's thesis that at the moment it is a debatable conclusion about the relationship between the results obtained and the magnitude of the upcoming event is not without grounds. The authors believe that such a relationship can be revealed in the course of further studies by comparing the total number of detected precursors and the magnitude of the subsequent earthquake. At the moment, it can be argued that, taking into account the logic and static meaning of the algorithm used, the increase in the number of considered precursors with a sufficiently high probability can be explained by the intensification of the process of "final preparation" of the approaching earthquake.

The corresponding clarification of the text is carried out in the "Discussion of the results" section of this article.

- **What is the IAGA code of the magnetic observatory situated in Katsiveli? I couldn't find it on intermagnet.org**

  **Answer:**

  It's SIM (Simeiz), number 33402.

- **Why the analysis on the geomagnetic observations doesn't consider the solar forcing? Magnetic field variation arises from current systems caused by solar radiation changes. Other irregular current systems produce magnetic field changes caused by the interaction of the solar wind with the magnetosphere, leading to the so-called solar wind-magnetosphere coupling. Geomagnetic data are influenced by external forcing, how can you justify the link with the earthquake without considering the external forcing at all?**

  **Answer:**

  Geomagnetic data are influenced by external forcing. The analysis on the geomagnetic observations consider the solar forcing. In Fig.4 shows the K-index of solar radiation changes during the period 00:00:00 on 24.06.2021 – 23:59:59 on 26.06.2021. As can be seen from the chart, the K-index was in the quiet (green) zone.

[Figure]

  The corresponding clarification of the text is carried out in the "Discussion of the results" section of this article.

- **In table 1, the values of intervals from the time of realization of «graphic precursors» to the moment of the beginning of the earthquake are reported, and they are different for each component of the field. However, the authors showed that the average on 6 hours is similar. However, what is the reason and the meaning of this average? Why it should be of some importance? Furthermore, how can be justified that these "graphic precursors" appear at different times and in different number on the 3 components of the geomagnetic field?**

**Answer:**

The closeness of the average values of the considered time intervals corresponding to different components of the magnetic field is an argument in favor of the stability of the applied approach. Which in turn makes it possible to speak about the objectivity of the results obtained.

We also point out that the different time of occurrence of precursors in the analysis of the statistics of different components of the magnetic field can be explained by different amplitudes of the random process $x_2(t)$ for each of the components of this field. Such spatial anisotropy can be associated, in particular, with the existence of distinguished directions for the development of cracking in the zone of preparation of a seismic event. The consequence of this difference in amplitudes is a different degree of averaging of fluctuations in the probability density of the background noise for different components of the magnetic field. This effect manifests itself both in the noncoinciding form of the $L(n)$ curves corresponding to different components of the magnetic field, and in different times and in the number of detected precursors. In this case, the indicated proximity of the average values of the indicated time intervals corresponding to various components of the magnetic field can be explained by a synchronous change in the properties of the statistics of all magnetic field components at the start and/or end of the next stage of the "final preparation" of the approaching event. Thus, such a difference in the time of registration of individual precursors under consideration for different field components, in combination with a small relative difference in the corresponding average values, is quite consistent with the logic of the calculations.

The relevant text has been added to the "Discussion of the results" section of this article.

- **The comparison with the Borok magnetometer data for the Z-component is not so good. Following the authors criteria, the 6 hours average of intervals from the time of realization of «graphic precursors» to the moment of the beginning of the earthquake is equal to 102 which is not similar to the value of the corresponding component at the Katsiveli test site (246).**

**Answer:**

Taking into account this recommendation of the reviewer, the authors exclude from the text of the revised version of the article all comparisons with the data of the indicated magnetometer.

- **Lines 100-101: Why the duration of the channel existence must be at least 150 minutes? This and other constraints (for example the one at line 120) are not clear.**

**Answer:**

This condition, as well as all other restrictions specified in the article (see, in particular, (4) - (8)) are introduced in order to minimize the number of linear structures under consideration in the range from several units to about eight to ten. In this case, the conditions for analyzing the degree of concentration of the considered precursors over limited time intervals are significantly simplified. As follows from the results of this article, such a concentration with a high probability precedes the onset of an earthquake.

The relevant text has been added to the "Data and analysis of the properties of the functional L(n) constructed from the measurements of the magnetic field" section of this article.

- **Line 111: Maybe there are some typos otherwise the sentence is not clear.**
- **Lines 184-185: the verb is missing in this sentence.**

**Answer:**

We corrected the text.

- **Line 147-148: Please justify the sentence: "The fact of the existence of such channels can be interpreted as the emergence of a set of deterministic processes immediately before an earthquake."**
- **Line 152-153: Please justify the sentence: "Therefore, we will consider the moment of the indicated fifth testing as the time of realization of one of the types of "graphic precursors" of an impending earthquake".**

**Answer:**

According to the statistical meaning of the functional $L(n)$, its maxima and minima with high probability correspond to the smallest and, accordingly, the largest level of variations of the random process $x_2(t)$, independent or weakly dependent on background noise, as applied to a local sequence of M implementation segments, see (2) that precede this point in time. The article introduces the hypothesis that such an independent or quasi-independent process is determined by a set of phenomena associated with the fracture of lithospheric plates in the area of preparation for an approaching earthquake. (It is taken into account that there are no significant geomagnetic disturbances associated with solar events during the period under study.) Therefore, if this hypothesis is true, the maxima and minima of the dependence $L(n)$ correspond to the minimum and, accordingly, the maximum values of the level of compression of lithospheric plates in the region of the future hypocenter.

Therefore, the appearance of a channel means the appearance of boundaries determined by deterministic or quasi-deterministic functions of time, within which the process of oscillations of the corresponding seismic pressure occurs. Similarly, the formation of a sliding boundary means the existence of such a limit, limiting from above or below the values of the specified pressure. It is in this sense that the term about the occurrence of deterministic phenomena used in this article should be considered.

We also note that, taking into account the definition and properties of the functional $L(n)$, the linear form of the boundaries of the indicated graphical objects, whether they be channels or sliding boundaries, can correspond to the hyperbolic form of quasi-deterministic time dependences that determine the boundaries of seismic pressure fluctuations during the "final preparation" period approaching earthquake. The validity of this assertion will be investigated in future works.

Taking into account the foregoing, let us additionally explain the physical meaning of the chosen criteria for registering the precursors of an approaching earthquake. The fifth test of the curve $L(n)$ of the boundaries of any of the channels means either a double "internal" oscillation within the boundaries of this channel (with the possibility of going beyond its limits at the point of the fifth test, see channel *5–6* in Fig. 1). This means a double approach of the seismic pressure level to some time-varying critical level, which is determined by a quasi-deterministic dependence, with the possibility of its "breakthrough" in the end. Or such a five-time test means from four to five consecutive "attempts to break through" (either only from above, or only from below) both boundaries of a given channel (see channel *1–2* in Fig. 1), also with the possibility of going beyond its limits during the last test . The physical meaning of such behavior of the $L(n)$ curve is reduced to five successive approaches of the seismic squeezing level to a certain critical level, which is also determined by a quasi-deterministic function of time. As follows from Volvach et al. 2022a, in both cases, such phenomena with a fairly high probability take place at the "final preparation" stage of the forthcoming earthquake.

The physical meaning of the criterion for registering precursors after the fourth test of the $L(n)$ curve of the sliding boundary line is similar. It comes down to identifying at least four consecutive facts of reaching a certain time-varying critical level, which is determined by a quasi-deterministic dependence, with the possibility of its "pushing through" as a result. Also according to Volvach et al. 2022a, such effects with a fairly high probability occur in a fairly short, on average, about several hours, time interval before the onset of the upcoming earthquake.

The corresponding clarification of the text is carried out in the "Discussion of the results" section of this article.

In the article, an additional tightening of the applied methodology was introduced in the form of the requirement that any control point should be horizontally separated from any other point of the curve $L(n)$ for a time interval $\Delta n \geq 35 \ min$.

At the same time, for Fig. 3, some marked linear objects and their corresponding moments of precursor registration were corrected.
* * *

---

## Author Comment (AC4)

[revised manuscript text omitted]
. At the same time, an additional condition is introduced that any point considered as the main extremum must be horizontally removed from any other point of the $L(n)$ curve by at least 35 minutes. Here and below, unless the opposite condition is set, small-scale variations in the values of $L(n)$ are not taken into account. This term will denote fluctuations that are much smaller in modulus than the right side of (4) and, at the same time, are small in amplitude and duration compared to the variation of the functional $L(n)$ in the region of the corresponding extremum. Thus, local trends are rather large quasi-rectilinear segments of the $L(n)$ dependence, the boundary points of which are extrema, horizontally separated by a given minimum distance from any other points of the curve $L(n)$. 
[revised manuscript text omitted]

This condition, as well as all other restrictions specified in the article (see, in particular, (4) - (8)) are introduced in order to minimize the number of linear structures under consideration in the range from several units to about eight to ten. In this case, the conditions for analyzing the degree of concentration of the considered precursors over limited time intervals are significantly simplified. As follows from the results of this article, the beginning of the considered earthquake is preceded by a sharp increase in the concentration of these precursors over the last few hours.

Further, when analyzing the process of "final preparation" of a seismic event, as well as when determining the average duration of the interval from the moment of registration of a precursor to the onset of an earthquake, we will take into account only those precursors that are separated from the time of its onset by no more than an interval

$$T_{max} = 720 \; min \tag{8}$$

Therefore,  the time interval from point $v$ in Fig. 1 и два подобных интервала от первых двух предвестников

на Fig. 3 until the moment of the seismic event is not taken into account.

In Fig. 2 shows the dependence $L(n)$ for the E-component of the magnetic field (data from the magnetometer in Katsiveli).

Here, channels *1–2*, *3–4* and *5–6* are drawn, for which the angles $\Delta\alpha < \Delta\tilde{\alpha} \ll 1°$ (see (5) and (6)). Segments $a-b, i-j,$

[revised manuscript text omitted]

7. Note that Volvach et al. 2022a,b,c,d; Volvach et al. 2023a, using an approach similar to that used in this article, precursors for 37 earthquakes are considered. For 26 of them (with the exception of aftershocks that are not far removed in time from the previous event), almost all of the considered precursors are recorded before their "own" earthquake at a time interval of no more than 20–25% of the total considered measurement period $\Delta t_{glob}$. Outside a given period of time, the number of such phenomena is much less than within it. Moreover, in the absolute majority of the cases under consideration, this interval is on the order of 5–10% or less of $\Delta t\_glob$. The same effect also takes place in relation to the present article, which indicates the stability of the results of the method used. This fact means a high probability that the statistical effects under consideration are related precisely to the process of the "final preparation" of the approaching earthquake.

8. As follows from this article, as well as Volvach et al. 2022a,b; Volvach et al. 2023a, if at least 3 considered precursors occur within an interval of 300 minutes or more, according to the measurement statistics of at least one of the magnetic field components, the probability that an earthquake will occur within an interval of about an hour after the registration of the last specified precursor is very high. We emphasize that this conclusion is obviously preliminary, subject to verification in the course of further studies and relating to the case of a total duration of the analyzed measurement period of the order of two days.

9. Note that the relative closeness of the mean values of the considered time intervals corresponding to different components of the magnetic field (see Table 1) is an argument in favor of the stability of the applied approach. Which in turn makes it possible to speak about the objectivity of the results obtained.

We also point out that the different time of occurrence of precursors in the analysis of the statistics of different components of the magnetic field can be explained by different amplitudes of the random process $x_2(t)$ for each of the components of this field. Such spatial anisotropy can be associated, in particular, with the existence of distinguished directions for the development of cracking in the zone of preparation of a seismic event. The consequence of this difference in amplitudes is a different degree of averaging of fluctuations in the probability density of the background noise for different components of the magnetic field. This effect manifests itself both in the noncoinciding form of the $L(n)$ curves corresponding to different components of the magnetic field, and in different times and in the number of detected precursors. In this case, the indicated proximity of the average values of the indicated time intervals corresponding to various components of the magnetic field can be explained by a synchronous change in the properties of the statistics of all components of the magnetic field, the moments of the beginning and/or completion of the next stage of the "final preparation" of the approaching event. Thus, such a difference in the time of registration of individual precursors under consideration for different field components, in combination with a small relative difference in the corresponding average values, is quite consistent with the logic of the calculations.

10. We also note that, according to the statistical meaning of the functional $L(n)$, its maxima and minima with high probability correspond to the smallest and, accordingly, the largest level of variations of the random process $x_2(t)$, independent or weakly dependent on background noise, as applied to a local sequence from $M$ segments of realization, see (2), which precede the given moment of time. The article introduces the hypothesis that such an independent or quasi-independent process is determined by a set of phenomena associated with the fracture of lithospheric plates in the area of preparation for an approaching earthquake. (It is taken into account that there are no significant geomagnetic disturbances associated with solar events during the period under study.) Therefore, if this hypothesis is true, the maxima and minima of the dependence $L(n)$ correspond to the minimum and, accordingly, the maximum values of the level of compression of lithospheric plates in the region of the future hypocenter.

Therefore, the appearance of a channel means the appearance of boundaries determined by deterministic or quasi-deterministic functions of time, within which the process of oscillations of the corresponding seismic pressure occurs. Similarly, the formation of a sliding boundary means the existence of such a limit, limiting from above or below the values of the specified pressure. It is in this sense that the term about the occurrence of deterministic phenomena used in this article should be considered.

We also note that, taking into account the definition and properties of the functional $L(n)$, the linear form of the boundaries of the indicated graphical objects, whether they be channels or sliding boundaries, can correspond to the hyperbolic form of quasi-deterministic time dependences that determine the boundaries of seismic pressure fluctuations during the "final preparation" period approaching earthquake. The validity of this assertion will be investigated in future works.

Taking into account the foregoing, let us additionally explain the physical meaning of the chosen criteria for registering the precursors of an approaching earthquake. The fifth test of the curve $L(n)$ of the boundaries of any of the channels means either a double "internal" oscillation within the boundaries of this channel (with the possibility of going beyond its limits at the point of the fifth test, see channel *5–6* in Fig. 1). This means a double approach of the seismic pressure level to some time-varying critical level, which is determined by a quasi-deterministic dependence, with the possibility of its "breakthrough" in the end. Or such a five-time test means from four to five consecutive "attempts to break through" (either only from above, or only from below) both boundaries of a given channel (see channel *1–2* in Fig. 1), also with the possibility of going beyond its limits during the last test . The physical meaning of such behavior of the $L(n)$ curve is reduced to five successive approaches of the seismic squeezing level to a certain critical level, which is also determined by a quasi-deterministic function of time. As follows from Volvach et al. 2022a, in both cases, such phenomena with a fairly high probability take place at the "final preparation" stage of the forthcoming earthquake.

The physical meaning of the criterion for registering precursors after the fourth test of the $L(n)$ curve of the sliding boundary line is similar. It comes down to identifying at least four consecutive facts of reaching a certain time-varying critical level, which is determined by a quasi-deterministic dependence, with the possibility of its "pushing through" as a result. Also according to Volvach et al. 2022a, such effects with a fairly high probability occur in a fairly short, on average, about several hours, time interval before the onset of the upcoming earthquake.

**5 Conclusions**

[revised manuscript text omitted]

**Kp index (curve 4).**